# Hunting Site Behaviour of Sympatric Common Buzzard *Buteo buteo* and Rough-Legged Buzzard *Buteo lagopus* on Their Wintering Grounds

**DOI:** 10.3390/ani13172801

**Published:** 2023-09-04

**Authors:** Paweł Cieśluk, Maciej Cmoch, Zbigniew Kasprzykowski

**Affiliations:** Faculty of Science, Siedlce University of Natural Sciences and Humanities, Prusa 14, 08-110 Siedlce, Poland; maciejcmoch@interia.pl (M.C.); zbigniew.kasprzykowski@uph.edu.pl (Z.K.)

**Keywords:** birds of prey, foraging site, interspecies competition, non-breeding season, hunting time, weather parameters

## Abstract

**Simple Summary:**

We compared the foraging techniques of Common Buzzards and Rough-legged Buzzards on their wintering grounds in east-central Poland. Both buzzard species spent the most time standing on the ground, less perching on trees and even less perching on fence posts. The difference in the hunting behaviors of the two species is associated with the use of small fence posts around pastures as hunting sites, which were conspicuously avoided by the Rough-legged Buzzards. Snow cover was the only weather factor in both buzzard species that affected foraging behavior and possibly intensified interspecific competition.

**Abstract:**

Birds wintering in the northern Palearctic compensate for substantial energy losses and prepare for a food deficit in winter by adjusting their foraging behavior. Apart from weather conditions, interspecific competition also drives hunting strategies. To describe this phenomenon, we observed the behavior of two sympatrically wintering raptor species: the Common Buzzard and the Rough-legged Buzzard. The study was carried out in east-central Poland during four seasons on a study plot where the densities of both species were high. Interspecific differences were detected in the use of available hunting sites. Rough-legged Buzzards conspicuously avoided using fence posts for scanning the surroundings and spent the most time standing on the ground. Common Buzzards more often used trees for this purpose when the snow cover was thick. Thicker snow cover resulted in fewer attempted attacks on prey in both species and caused Common Buzzards to change their hunting sites less frequently. The study also showed that the more often a bird changed its hunting site, the greater the number of attempted attacks. The outcome is that the ultimate effectiveness of hunting is mediated by the overview of the foraging area from different heights and perspectives, not by the type of hunting site. Snow cover was the most important factor in modifying foraging behavior and possibly intensifying interspecific competition.

## 1. Introduction

Winter is a difficult time in the life of many organisms. In addition to the negative impact of low temperatures, the lack or limited availability of food also leads to considerable energy losses in animals and possibly their death [1,2,3,4]. This is why an adequate supply of energy in the form of high-calorie food [5,6,7] or sufficient amounts of other food [8] is so important during this season. Vertebrates wintering in the northern regions of the Northern Hemisphere adjust their foraging behavior accordingly to compensate for great losses of energy and/or to prepare for winter food deficiency. A frequent behavioral adaptation to the hardships of winter life is to store food [9], which is common in various species of mammals [10,11,12]. Birds likewise hoard food in winter, storing their stocks in tree hollows and nest boxes [13,14,15,16]. Other means of surviving the winter involve accumulating energy reserves in the form of fat tissue in summer and early autumn when food is still plentiful [17]. Losses of energy in winter can also be avoided by reducing total diurnal physical activity [18]. During this time, some birds of prey employ the energy-saving sit-and-wait hunting strategy [1,3,19,20,21,22].

Another factor crucial for the winter survival of birds is the weather. Heavy snowfall in mid-winter immediately reduces the population, which is associated with a mosaic-like farming landscape [23]. For raptors living in such a landscape, thick snow hampers or prevents the detection and pursuit of rodents [1,24]. Common Buzzards *Buteo buteo* from central Europe react to low temperatures by migrating southwards [25,26]. At the same time, there is an influx of Rough-legged Buzzards *Buteo lagopus* from the north, for which central Europe is a warmer region than central and southern Scandinavia [27]. In response to deteriorating weather conditions, raptors reduce their energy expenditure, particularly when prey items are in short supply [28], and resort to the sit-and-wait hunting strategy [1,3,19,20,21]. Sit-and-wait is obviously less energy-demanding than other foraging techniques, but it is also time-consuming [19]. For both European buzzard species, this strategy means using various kinds of hunting sites in farmland, including man-made ones, especially if natural ones are not available [19,20,29]. Common Buzzards can adjust their choice of hunting sites to weather conditions [30]: after heavy snowfall, they congregate along roads and railway lines, where hunting opportunities are better, and they can find carrion [19,31].

Interspecific competition between birds in which feeding niches overlap is another factor that can affect winter survival [32,33]. Closely related species of animals often share resources such as space and food to minimize competition and enable the divergence of their ecological niches [34,35]. The two species of wintering buzzards, which employ the same hunting strategy for the same source of food, provide a good example of these processes [36]. 

We compared the foraging techniques of Common Buzzards and Rough-legged Buzzards on their wintering grounds in east-central Poland. Both species employed an energy-conserving sit-and-wait strategy [1] and fed on small mammals, mostly on the Common Vole *Microtus arvalis* [37]. The main aims of this study were (1) to investigate which sites were used for hunting and (2) to determine factors that mediated the number of attacks on prey in both buzzard species. We hypothesized that the time spent at each type of hunting site would differ between Rough-legged Buzzards and Common Buzzards. We also expected that species and time spent on different hunting sites, as well as the number of changes in hunting sites and weather conditions, would affect the number of attacks on prey. Knowledge of the differences in hunting techniques and the factors that improve foraging success between these two buzzard species may contribute to a better understanding of how competition is reduced and how niche differentiation strategies develop in morphologically and ecologically similar species [35,38,39].

## 2. Materials and Methods

### 2.1. Study Plot

The study plot (area 18.9 km^2^) was situated in the upper valley of the River Liwiec in central Poland in a complex of hay meadows and pastures of diverse humidity, criss-crossed by numerous drainage ditches (Figure 1). Fencing posts and trees provided potential hunting sites for buzzards. Most of these posts were located in meadows, which covered about 94.5% of the study plot [40], with the average area of a fenced meadow being about 1 ha (range 0.2–2.5 ha) [41]. Thus, fencing posts were fairly evenly distributed throughout the plot. Single trees, as well as clumps of trees (willow shrubs), grew along the drainage ditches, so their distribution was also even. This particular area supports the highest densities of wintering buzzards in east-central Poland. The density of Common Buzzards in the study plot was 6.84 ind./km^2^ compared with an average of 1.32 ind./km^2^ recorded in central-eastern Poland [40]. The densities of Rough-legged Buzzard within the study plot were also high (1.16 ind./km^2^) compared to other areas in east-central Poland (0.59 ind./km^2^) [40,42].

### 2.2. Data Collection

Observations were carried out in this study plot during four winter seasons—2007/2008, 2008/2009, 2011/2012, and 2012/2013—from the first days of November to the end of February. Birds were monitored at roughly two-week intervals on days without rain or snow. Around 9 to 14 visits were made in each season, a total of 44. The birds were always counted during the same hours of the day (07:30–13:30). 

Behavioral observations were made from vantage points with good visibility using a 20–60 × 100 spotting scope. Individual birds were tracked for a minimum of 10 min and a maximum of 30 min. If a buzzard had disappeared before 10 min passed, its observation ceased, and another bird was chosen. Each visit involved an average of 65 min of observations (range 20–130 min.). The analyses treated each 10-min sequence as a separate sample. A total of 1140 min (114 sequences) was spent observing Common Buzzards and 1610 min (161 sequences) on Rough-legged Buzzards. 

The time spent on each hunting site was recorded to within one second using a dictaphone. Additional information was also assigned to each recording: the date and time of the recording and snow thickness. Mean daily temperatures (°C) and mean wind speeds (km/h) were obtained from the Siedlce weather station (52°25′ N; 22°26′ E), situated about 17 kilometers away. The thickness of the snow cover was measured at 4 randomly selected sites within the study plot during each observation. Because both buzzard species spent most of their time at their hunting sites (Common Buzzard—96.5% of the time, Rough-legged Buzzard—91.4%), only this type of activity was analyzed. In accordance with the recommendations of Bohall and Collopy [43], Wuczyński [20], and Bylicka et al. [21], which we slightly modified, three types of hunting sites were considered: trees, fence posts, and the ground. All the sites were classified into these three categories. 

### 2.3. Statistical Analyses

A generalized linear mixed model (GLMM) with logit link function and binomial error variance was applied to compare the times spent on the three types of hunting sites (tree, fence post, and ground) by the two buzzard species (Table 1). The dependent variable was the species of the compared pair (binomial variable: 0–Common Buzzard, 1–Rough-legged Buzzard). A second generalized linear mixed model with Poisson distribution and logit link function was used to analyze the number of attacks by the buzzards. The time spent on the three types of hunting sites, the number of changes of hunting sites, the mean temperature, mean wind speed, and snow cover were treated as numerical predictors. The third model, where the dependent variable was the number of hunting site changes, analyzed three weather parameters (mean temperature, mean wind speed, and snow cover), together with the interaction between species and the number of changes of hunting site, using a Poisson distribution and logit link function. The birds were not individually marked, so some may have been recorded more than once. The inclusion of the observation number as a random effect in the two models addressed the question of pseudoreplication. The second random factor was the winter season. The differences in snow cover between the hunting sites were tested using Tukey’s post-hoc test. Student’s *t*-test was applied to assess the differences in the number of changes of hunting sites between two snow cover categories (present and absent). Prior to the parametric analyses (Student’s *t*-test), all the dependent variables were log (x + 1) transformed to obtain a normal distribution. All the statistical analyses were performed using R software 4.2.2 [44].

## 3. Results

Both buzzard species spent most of their time standing on the ground, less perching on trees, and the least perching on fence posts (Figure 2). Only the difference between perching time on fence posts was significant between species (Table 2). Rough-legged Buzzards conspicuously avoided such hunting sites (Figure 2). 

The number of attacks on prey was influenced by two predictors: the number of hunting site changes and the snow cover (Table 3). The number of changes in the hunting site positively affected the number of attacks. The thickness of snow cover was the only weather factor that significantly and negatively influenced the number of attacks on prey. There were no differences between the species in either the number of attacks or the time spent at particular hunting sites. However, when the influence of snow cover was analyzed together with species, there was a difference in this weather parameter between the use of hunting sites by Common Buzzards and Rough-legged Buzzards (Table 4). Common Buzzards perched on trees more frequently than on the other two types of hunting sites when the snow cover was significantly thicker (Tukey’s post-hoc test, *p* < 0.001 for both comparisons, Figure 3). There was no difference in the snow cover between the use of the ground or fence posts for hunting sites (Tukey’s post-hoc test, *p* = 0.881). 

The number of changes in the hunting site depended on the snow cover analyzed as a separate variable and on the interaction of the snow cover and the species (Table 4). Common Buzzards changed their foraging sites less often in the presence of snow cover than when there was no snow, which was a significant difference (Student’s *t*-test, *t* = 3.49, *p* < 0.001, df = 111). No influence of snow cover on the number of changes in hunting sites was found for Rough-legged Buzzards (Student’s *t*-test, *t* = 1.09, *p* = 0.277, df = 158).

## 4. Discussion

Our study showed that the hunting time differed between the two buzzard species only for fence posts. Snow cover, but no other weather conditions, influenced the number of attacks on prey and the number of changes in hunting sites. However, the behavior of Common Buzzards appeared to be more dependent on the occurrence of snow cover than the hunting techniques used by Rough-legged Buzzards.

A significant difference in hunting behavior between the two species is the use of small fence posts in pastures as hunting sites, which the Rough-legged Buzzards conspicuously avoided. This may occur because this species repeats its breeding-ground foraging strategies in its wintering areas. Potapov [45] draws attention to the fact that across the Rough-legged Buzzard’s breeding area, there are practically no signs of human activity. This might explain why Rough-legged Buzzards did not use man-made objects as hunting sites to the extent that Common Buzzards did. Unfortunately, determining the hunting success of the two buzzard species was difficult, so it was not possible to analyze this factor in relation to the type of hunting site.

The preference of both species of buzzards for standing on the ground in our study plot may be caused by several factors that result indirectly from its grassland character. Firstly, low vegetation in hay meadows and pastures during winter favors this hunting technique [20,21]. Secondly, hay meadow habitats support a large abundance of Common Voles *Microtus arvalis*, the main prey of the buzzards’ winter diet [37]. Standing on the ground may also reduce the effects of inter and intra-specific competition since a bird hunting on the ground immediately swallows captured rodents whole without tearing them into pieces, thus avoiding the risk of being robbed [20]. Foraging on the ground is an energetically profitable means of acquiring food, especially when food is plentiful. In winter, every return flight to a perch is energetically costly [20]. Müller et al. [46] and Gamauf [47] described foraging on the ground as an exceptionally effective method of hunting rodents, particularly suitable for young, inexperienced Common Buzzards and habitats with the best food supply. Dare [48] notes that foraging on the ground, especially in autumn and winter, is a common method used by Common Buzzards when hunting for beetles, earthworms, and other ground-dwelling invertebrates. 

We found that both species performed a similar number of attacks on prey. This is explained by their use of similar foraging techniques. Unlike species actively seeking prey, such as the Kestrel *Falco tinnunculus* and Hen Harrier *Circus cyaneus*, both buzzard species ambush their victims from a sit-and-wait position [20,21,31]. Nonetheless, a far more important factor mediating the numbers of attempted attacks on prey by both species appeared to be individual decisions, manifested by the number of changes of hunting site per time unit. Our study showed that buzzards were able to improve their hunting success by moving from one hunting site to another close by, thus increasing the controlled area [31]. On the other hand, we found that the time a buzzard spent on hunting sites of different types was not related to the number of attempted attacks. Therefore, it seems that the kind of hunting site is less important than the hunting buzzard’s view of its foraging area from different heights and perspectives. Our study also showed that the thicker the snow cover, the fewer attacks were attempted on prey in both buzzard species. A further consequence of persistent thick snow cover is the decrease in numbers of both buzzard species on the wintering grounds and their south-westward migration [23,24,40,47,49]. Smaller numbers of Common Buzzards in the natural open habitats, associated with the thicker snow cover, could also result from local movements toward roads, where Rough-legged Buzzards are not recorded [25,30,31]. 

Our analysis also showed that Common Buzzards changed their hunting sites less often if the ground was covered with snow. The energy expense necessary for moving to higher hunting sites, giving a better view of potential victims on the snow, is thus reduced [1]. No such relationship was found in the Rough-legged Buzzard. This species probably tolerates a thicker snow cover at its wintering grounds and will, therefore, not alter its behavior. More frequent use of trees as hunting sites by Common Buzzards when the snow was thicker further confirms that pattern. This type of behavior might be interpreted as the effect of interspecific competition when deteriorating weather conditions drive arrivals of larger numbers of Rough-legged Buzzards from the regions to the north of our study plot [42,50], whereas, in spring, the return migration of Rough-legged Buzzards is synchronized with the northwards progression of snowmelt [51,52]. 

## 5. Conclusions

We showed that the Rough-legged Buzzard seems to be more conservative in its use of hunting sites and less likely to change its foraging behavior when weather conditions deteriorate. In contrast, the Common Buzzard varies its hunting techniques in response to the appearance of snow cover, attempting fewer attacks on prey and using trees as hunting sites more frequently. Under conditions of reduced food availability, changes in hunting strategies may result from increased interspecific competition. Given the Rough-legged Buzzard’s larger body size, the Common Buzzard may have to resort to less effective hunting strategies. However, this is a supposition that requires further behavioral studies on wintering grounds used by both species. 

## Figures and Tables

**Figure 1 animals-13-02801-f001:**
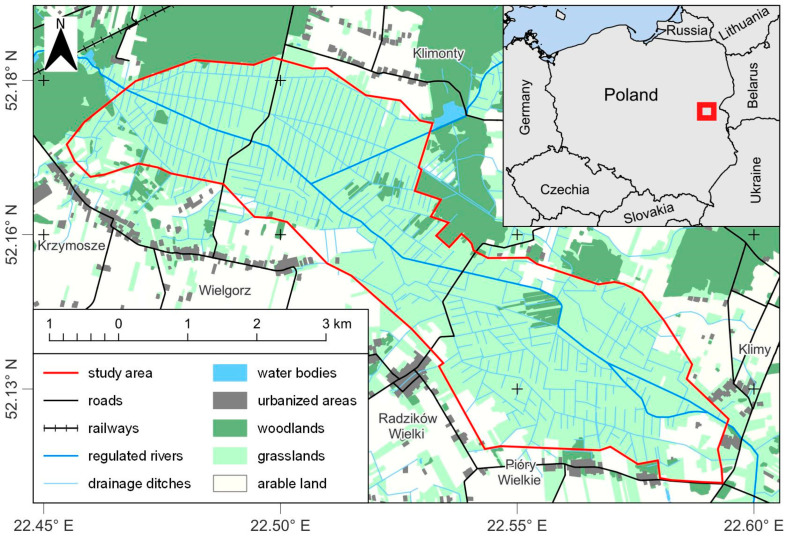
Study map.

**Figure 2 animals-13-02801-f002:**
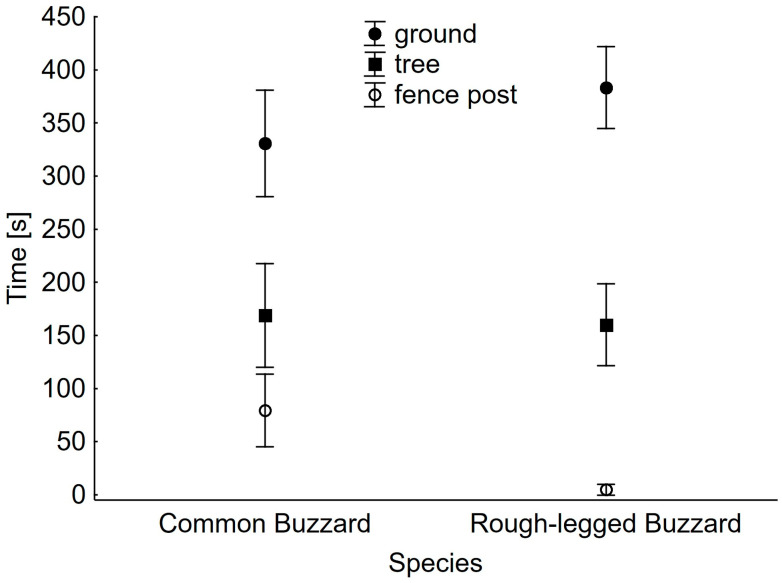
Mean time spent by Common and Rough-legged Buzzards hunting on three types of sites. Whiskers indicate 95% confidence limits.

**Figure 3 animals-13-02801-f003:**
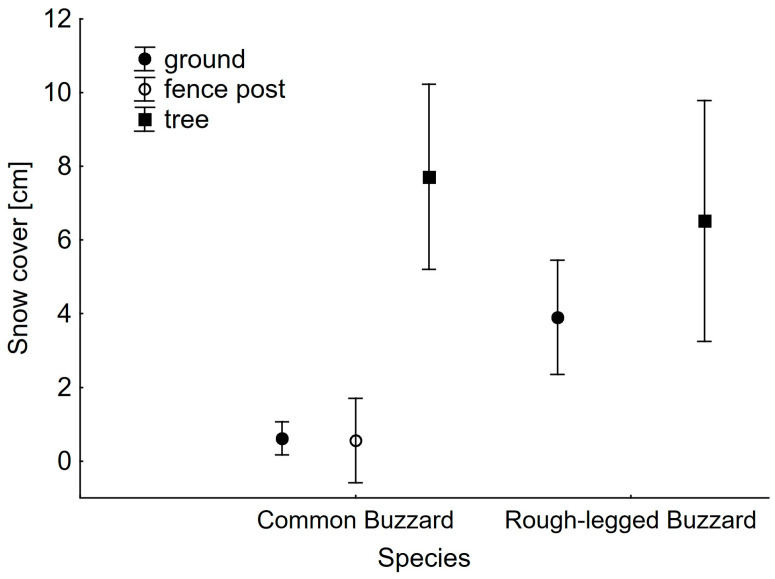
Mean snow cover during the use of hunting sites by Common and Rough-legged Buzzards. Whiskers indicate 95% confidence limits.

**Table 1 animals-13-02801-t001:** Factors used to explain differences in hunting sites between Common Buzzard and Rough-legged Buzzard and the number of attacks by the buzzards.

Code	Description
TimeTree	Time spent on a tree (s)
TimePost	Time spent on a fence post (s)
TimeGround	Time spent on the ground (s)
Change	Number of changes in hunting sites
Temp	Mean temperature (°C)
Wind	Mean wind speed (km/h)
Snow	Thickness of s now cover (cm)

**Table 2 animals-13-02801-t002:** Results of a binomial generalized linear mixed model comparing the times spent on the three types of hunting sites between Common Buzzard and Rough-legged Buzzard.

Variable	Estimate	Standard Deviation	Z-Value	*p*-Value
TimeTree	−0.008	0.018	−0.443	0.658
TimePost	−0.138	0.055	−2.505	0.012
TimeGround	−0.008	0.019	−0.410	0.682

**Table 3 animals-13-02801-t003:** Results of a generalized linear mixed model testing the influence of different factors on the number of attacks on prey by the Common Buzzard and Rough-legged Buzzard.

Factor	Estimate	Standard Deviation	Z-Value	*p*-Value
Species	0.386	0.211	1.831	0.067
TimeTree	0.001	0.001	−0.319	0.750
TimePost	0.001	0.001	0.605	0.545
TimeGround	0.000	0.001	0.398	0.690
Change	0.409	0.040	10.273	<0.001
Temp	0.029	0.022	1.355	0.176
Wind	0.012	0.023	0.517	0.605
Snow	−0.038	0.017	−2.289	0.022

**Table 4 animals-13-02801-t004:** Results of a generalized linear mixed model testing the influence of different factors on the number of hunting site changes. Interactions between factors are marked with ×.

Factor	Estimate	Standard Deviation	Z-Value	*p*-Value
Species	0.042	0.499	0.085	0.932
Snow	−0.101	0.031	−3.265	0.001
Temp	−0.016	0.037	−0.426	0.670
Wind	0.022	0.033	0.668	0.504
Species × Snow	0.093	0.033	2.806	0.005
Species × Temp	0.053	0.043	1.238	0.216
Species × Wind	−0.024	0.040	−0.602	0.547

## Data Availability

Datasets supporting the reported results can be found at the Mendeley Data repository: https://data.mendeley.com/datasets/tcs5vsdmnm/1 (accessed on 4 July 2023).

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
