# Peer review of "Hunting Site Behaviour of Sympatric Common Buzzard Buteo buteo and Rough-Legged Buzzard Buteo lagopus on Their Wintering Grounds"

_animals, 2023, doi:10.3390/ani13172801_

Round 1
Reviewer 1 Report
Review Perching behavior Buzzards on their wintering grounds
A nice field study on perching behavior of Common and Rough-legged Buzzards in eastern Poland during winter. Data was collected from vantage points with good visibilities, ranging 20-130 minutes with observing individual birds for 10 minutes, completely random. All observations of study species were noted. This offers a good study design to answer the objectives correctly.
Below detailed other comments:
Ln 55-56: Do authors mean harsh weather movements during winter or spring/autumn migration movements? In the first option would NE-Europe not be ‘too north’? Or do the authors mean that central/southern Scandinavian Buzzards move south? Exactly like Rough-legged Buzzards?
Ln 75-77: ‘Where they were sympatric’? As written now, I would as a reader, expect a difference when only one species is available. Or are we talking about density dependence behavior?
Ln 77: The main objective of this research
Ln 79-81: ‘Our hypothesis was that hunting techniques differed among species.’ Or actually not hunting techniques but perching time, since only ‘sit-and-wait’ strategies where observed (Ln 76). And why would you expect differences in perching times? Or do the authors mean that perching sites are different techniques? Please state more clear
Ln 82: which both factors, but see above remark
Ln 94-95: number of decimals must be the same
Ln 98-99: suggestion: ‘Within the study plot a density of 6.84 Common Buzzards/km2 was found, compare with an average of 1.32 birds recorded elsewhere in central-eastern Poland [41]’
Ln 100-101: not clear where 0.23 and 1.16 ind/km2 Rough-legged Buzzards are recorded. Within the study area? Why this range for this species and not for Buzzard?
Ln 122: ‘situated 17 km from the study plot’
Ln 131-149: Since GLMM are used for all three analyses, please concise this paragraph like something ‘GLMM with Posson distribution and logit link function was used to analyze: A, B and C’
Ln 158, fig 2: Please modify this graph as three high/low or maybe even better bar charts with SE per species next to each other. Current figure is not easy to read
Ln 156: replace ‘was statistically significant’ with ‘was significant between species’
Ln 156: table 2 is a bit too much. Data are shown in Fig 2 and result is in Ln 155-156, suggest to replace ‘(Table 2)’ with the statistical outcome showing test result, P-value and N, like the ANOVA results in Ln 171 and 172.
Ln 164-166: Don’t write like in a report or talk in an oral presentation, this is a peer-reviewed paper. Please rewrite 164-166
Ln >169: ?? You choose to build GLMMs with species as a categorical parameter. The result is thus presented in table 3, with Change and SnowCover as the significant parameters. Why would you add ANOVA for SnowCover? It’s already in your model. One other solution is already given by your third analysis: add a Species*Snow parameter. But be very careful with your interpretation. This paragraph as a whole should be rewritten
Ln 181, Fig3: would be nice if the chart of Fig2 and F3 are the same type. If you choose bar charts with SE in Fig2, please do so in Fig3. Same holds for the X-axis (or bird species first, as in Fig2; or perching first, as in Fig3)
Ln 185-190: Don’t mix up GLMM and ANOVA and pls rewrite the results first. What’s the outcome of GLMM (Snow and Species*Snow) and present the significant difference between the species as a t-test (as done). Fig 4 doesn’t add anything
Ln 194: Fig 4 can be removed, doesn’t add anything
Ln 217: ‘component’….? Would suggest to replace by prey(-item)
Ln 218: competition between who? Inter or Intra specific competition or both? Does density dependence not count as an argument for competition in this high density ‘buzzard’ study area?
Ln 218-220: A bit of an awkward sentence; how do ‘buzzards’ eat their prey when attacking from the other perches?
Ln 222: remove ‘very’ before costly. It could take up more energy compared to ground hunting (?). Any (literature) data proving this statement?
L
The structure (shortening sentences) and use of the English language needs considerable changes before publication.
Author Response
Manuscript “Perching behaviour of sympatric Common Buzzard Buteo buteo and Rough-legged Buzzard Buteo lagopus on their wintering grounds”
Responses and authors’ comments on the reviews
In accordance with the suggestions of the reviewers, we have made many amendments to the whole manuscript. The results, methods and discussion have been rewritten. In our opinion, the current version of the manuscript is now much clearer. We would like to thank both reviewers for their valuable comments. Below we describe the amendments in detail.
Response to Reviewer 1 Comments
A nice field study on perching behavior of Common and Rough-legged Buzzards in eastern Poland during winter. Data was collected from vantage points with good visibilities, ranging 20-130 minutes with observing individual birds for 10 minutes, completely random. All observations of study species were noted. This offers a good study design to answer the objectives correctly.
Below detailed other comments:
Ln 55-56: Do authors mean harsh weather movements during winter or spring/autumn migration movements? In the first option would NE-Europe not be ‘too north’? Or do the authors mean that central/southern Scandinavian Buzzards move south? Exactly like Rough-legged Buzzards?
The sentence has been changed to: “Common Buzzards Buteo buteo from central Europe react to low temperatures by migrating southwards”.
Ln 75-77: ‘Where they were sympatric’? As written now, I would as a reader, expect a difference when only one species is available. Or are we talking about density dependence behavior?
This phrase has been delated. Current form of sentence is: “Both species employed an energy-conserving sit-and-wait strategy [1] and fed on small mammals, mostly on the Common Vole Microtus arvalis”.
Ln 77: The main objective of this research
The phrase has been changed.
Ln 79-81: ‘Our hypothesis was that hunting techniques differed among species.’ Or actually not hunting techniques but perching time, since only ‘sit-and-wait’ strategies where observed (Ln 76). And why would you expect differences in perching times? Or do the authors mean that perching sites are different techniques? Please state more clear
We specified this sentence to: “We hypothesized that the time spent at each type of hunting sites would differ between Rough-legged Buzzards and Common Buzzards”. The explanation is in the last sentence of the introduction: “Knowledge of the differences in hunting techniques and the factors that improve foraging success between these two buzzard species may contribute to better understanding of how competition is reduced and niche differentiation strategies develop in morphologically and ecologically similar species”
Ln 82: which both factors, but see above remark
The sentence has been corrected: We also expected that species and time spent on different hunting sites, as well as the number of changes of hunting sites and weather conditions, would affect the number of attacks on prey.
Ln 94-95: number of decimals must be the same
It has been corrected.
Ln 98-99: suggestion: ‘Within the study plot a density of 6.84 Common Buzzards/km2 was found, compare with an average of 1.32 birds recorded elsewhere in central-eastern Poland [41]’
According to both reviewer comments we changed the sentence to: “The density of the Common Buzzard in the study plot was 6.84 ind./km2 compared with an average 1.32 ind./km2 recorded in central-eastern Poland”
Ln 100-101: not clear where 0.23 and 1.16 ind/km2 Rough-legged Buzzards are recorded. Within the study area? Why this range for this species and not for Buzzard?
The sentence has been corrected: The densities of Rough-legged Buzzard within the study plot was also high (1.16 ind./km2) compared to other areas (0.59 ind./km2)
Ln 122: ‘situated 17 km from the study plot’
It has been changed.
Ln 131-149: Since GLMM are used for all three analyses, please concise this paragraph like something ‘GLMM with Posson distribution and logit link function was used to analyze: A, B and C’
The first model was constructed with binomial error variance. The second and the third had Poisson distribution.
Ln 158, fig 2: Please modify this graph as three high/low or maybe even better bar charts with SE per species next to each other. Current figure is not easy to read
Figure 2 has been modified.
Ln 156: replace ‘was statistically significant’ with ‘was significant between species’
It has been replaced.
Ln 156: table 2 is a bit too much. Data are shown in Fig 2 and result is in Ln 155-156, suggest to replace ‘(Table 2)’ with the statistical outcome showing test result, P-value and N, like the ANOVA results in Ln 171 and 172.
Table 2 shows the results one of mixed model and consequently, as with other models, all statistics are presented.
Ln 164-166: Don’t write like in a report or talk in an oral presentation, this is a peer-reviewed paper. Please rewrite 164-166
The sentences had been changed.
Ln >169: ?? You choose to build GLMMs with species as a categorical parameter. The result is thus presented in table 3, with Change and SnowCover as the significant parameters. Why would you add ANOVA for SnowCover? It’s already in your model. One other solution is already given by your third analysis: add a Species*Snow parameter. But be very careful with your interpretation. This paragraph as a whole should be rewritten
According to suggestion we resigned from ANOVA. This paragraph has been rewritten.
Ln 181, Fig3: would be nice if the chart of Fig2 and F3 are the same type. If you choose bar charts with SE in Fig2, please do so in Fig3. Same holds for the X-axis (or bird species first, as in Fig2; or perching first, as in Fig3)
Figure 3 has been modified.
Ln 185-190: Don’t mix up GLMM and ANOVA and pls rewrite the results first. What’s the outcome of GLMM (Snow and Species*Snow) and present the significant difference between the species as a t-test (as done). Fig 4 doesn’t add anything
ANOVA analyses has been removed.
Ln 194: Fig 4 can be removed, doesn’t add anything
Figure 4 has been removed.
Ln 217: ‘component’….? Would suggest to replace by prey(-item)
It has been changed.
Ln 218: competition between who? Inter or Intra specific competition or both? Does density dependence not count as an argument for competition in this high density ‘buzzard’ study area?
The sentences has been corrected.
Ln 218-220: A bit of an awkward sentence; how do ‘buzzards’ eat their prey when attacking from the other perches?
The sentences has been corrected.
Ln 222: remove ‘very’ before costly. It could take up more energy compared to ground hunting (?). Any (literature) data proving this statement?
It has been corrected. There are not study compared energy costs between ground hunting and hunting from the perch in the literature.
The structure (shortening sentences) and use of the English language needs considerable changes before publication.
English language has been edited by a native speaker (Peter Senn and Sara Wild).
Reviewer 2 Report
The study provides an interesting and valuable example of the influence of the interspecific competition on behaviour of the two related species of buzzards wintering in the same area. The study is valuable, as it clearly shows that Common Buzzard can adjust their hunting behaviour to competition from the larger Rough-legged Buzzard, and that such changes appear when snow cover impedes hunting efficiency of the first species. The study provides good evidence of such patterns in a well designed statistical analyses. Discussion also explains the revealed patterns in an interesting way, in terms of avoiding competition. Thus, the paper is valuable contribution to the topic of effects of inter-specific competition on birds’ behaviour.
The one aspect which could be more emphasised in the discussion is whether such competition between the two species is usual and common pattern, and is even throughout winter. Or does the competition increase temporarily, with the influx of the Rough-legged Buzzards from the north, and occurrence of the snow cover at the same time?
The paper would greatly benefit from more careful and precise phrasing of the concepts by the authors. The results of statistical analyses in some places are described in a confusing or unclear wa, which impede understanding of what was analysed and what was actually the outcome in terms of birds' behaviour. Fortunately the reader can figure it from the tables. Also in discussion, more careful phrasing would make the author’s arguments easier to understand.
My main confusion with the terminology used throughout the paper is whether "the ground" can be considered as "the type of a perch". I guess that a perch, which is a pole or a twig etc., by defauls cannot also mean the ground. Maybe "the type of a perching site" would be a more fortunate phrase, or you can find a better phrase in the literature of the topic.
However, these flaws of the paper are small and can be easily fixed. I made more detailed comments in the attached manuscript.

The English terminology used is correct, and in some places the paper reads well, especially in introduction and discussion. But the paper has gramatical mistakes and phrasing is often unclear and confusing. The paper would greatly benefit from careful editing by the authors, supported with the use of language software, or by an English editor. I made some comments and suggested corrections in the attached manuscript.
Author Response
Manuscript “Perching behaviour of sympatric Common Buzzard Buteo buteo and Rough-legged Buzzard Buteo lagopus on their wintering grounds”
Responses and authors’ comments on the reviews
In accordance with the suggestions of the reviewers, we have made many amendments to the whole manuscript. The results, methods and discussion have been rewritten. In our opinion, the current version of the manuscript is now much clearer. We would like to thank both reviewers for their valuable comments. Below we describe the amendments in detail.
Response to Reviewer 2 Comments
The study provides an interesting and valuable example of the influence of the interspecific competition on behaviour of the two related species of buzzards wintering in the same area. The study is valuable, as it clearly shows that Common Buzzard can adjust their hunting behaviour to competition from the larger Rough-legged Buzzard, and that such changes appear when snow cover impedes hunting efficiency of the first species. The study provides good evidence of such patterns in a well designed statistical analyses. Discussion also explains the revealed patterns in an interesting way, in terms of avoiding competition. Thus, the paper is valuable contribution to the topic of effects of inter-specific competition on birds’ behaviour.
The one aspect which could be more emphasised in the discussion is whether such competition between the two species is usual and common pattern, and is even throughout winter. Or does the competition increase temporarily, with the influx of the Rough-legged Buzzards from the north, and occurrence of the snow cover at the same time?
Undoubtedly, this issue is very interesting to discuss. However, there is no data in the literature describing the occurrence of both species in a longer period of time and under changing weather conditions. Therefore, at the end of the conclusions, we put the following sentence: But this is a supposition that requires further behavioural studies on wintering grounds used by both species.
The paper would greatly benefit from more careful and precise phrasing of the concepts by the authors. The results of statistical analyses in some places are described in a confusing or unclear way, which impede understanding of what was analysed and what was actually the outcome in terms of birds' behaviour. Fortunately the reader can figure it from the tables. Also in discussion, more careful phrasing would make the author’s arguments easier to understand.
We hope that the corrections made based on the comments of all two reviewers will contribute to improving the paper.
My main confusion with the terminology used throughout the paper is whether "the ground" can be considered as "the type of a perch". I guess that a perch, which is a pole or a twig etc., by defauls cannot also mean the ground. Maybe "the type of a perching site" would be a more fortunate phrase, or you can find a better phrase in the literature of the topic.
We agree with reviewer that ground cannot be category of a perch. Therefore, we have used new better phrase “hunting sites” in the current version of ms.
However, these flaws of the paper are small and can be easily fixed. I made more detailed comments in the attached manuscript.
All corrections included in pdf version of the manuscript have been made.
Round 2
Reviewer 1 Report
Thank you for improving the language; the paper is very clear and ready for publication. Congrats!